# Fabrication routes *via* projection stereolithography for 3D-printing of microfluidic geometries for nucleic acid amplification

**Charalampos Tzivelekis**[1]*, **Pavlos Sgardelis**[1], **Kevin Waldron**[2], **Richard Whalley**[1], **Dehong Huo**[1], **Kenny Dalgarno**[1]

**1** School of Engineering, Newcastle University, Newcastle, United Kingdom, **2** Institute for Cell and Molecular Biosciences, Newcastle University, Newcastle, United Kingdom

* c.tzivelekis@newcastle.ac.uk

**Data Availability Statement:** All relevant data are within the manuscript and its Supporting Information files.

## Abstract

Digital Light Processing (DLP) stereolithography (SLA) as a high-resolution 3D printing process offers a low-cost alternative for prototyping of microfluidic geometries, compared to traditional clean-room and workshop-based methods. Here, we investigate DLP-SLA printing performance for the production of micro-chamber chip geometries suitable for Polymerase Chain Reaction (PCR), a key process in molecular diagnostics to amplify nucleic acid sequences. A DLP-SLA fabrication protocol for printed micro-chamber devices with monolithic micro-channels is developed and evaluated. Printed devices were post-processed with ultraviolet (UV) light and solvent baths to reduce PCR inhibiting residuals and further treated with silane coupling agents to passivate the surface, thereby limiting biomolecular adsorption occurences during the reaction. The printed devices were evaluated on a purpose-built infrared (IR) mediated PCR thermocycler. Amplification of 75 base pair long target sequences from genomic DNA templates on fluorosilane and glass modified chips produced amplicons consistent with the control reactions, unlike the non-silanized chips that produced faint or no amplicon. The results indicated good functionality of the IR thermocycler and good PCR compatibility of the printed and silanized SLA polymer. Based on the proposed methods, various microfluidic designs and ideas can be validated in-house at negligible costs without the requirement of tool manufacturing and workshop or clean-room access. Additionally, the versatile chemistry of 3D printing resins enables customised surface properties adding significant value to the printed prototypes. Considering the low setup and unit cost, design flexibility and flexible resin chemistries, DLP-SLA is anticipated to play a key role in future prototyping of microfluidics, particularly in the fields of research biology and molecular diagnostics. From a system point-of-view, the proposed method of thermocycling shows promise for portability and modular integration of funcitonalitites for diagnostic or research applications that utilize nucleic acid amplification technology.

**Funding:** This research was co-funded by the Engineering and Physical Sciences Research Council (EPSRC, https://epsrc.ukri.org/) (through award EP/L01534X/1), QuantuMDx Ltd (https://quantumdx.com/), and Newcastle University (https://www.ncl.ac.uk/). The EPSRC had no role in study design, data collection and analysis, decision to publish, or preparation of the manuscript. Newcastle University was involved in the study design, data collection, and analysis, the decision to publish and the preparation of the manuscript. QuantumDx was involved in the study design but did not contribute to the data collection, analysis, the decision to publish or the preparation of the manuscript.

**Competing interests:** The authors have read the journal's policy and have the following potential competing interests: QuantumDx had a role in the study design. This does not alter our adherence to PLOS ONE policies on sharing data and materials. There are no patents, products in development or marketed products associated with this research to declare.

## 1. Introduction

### 1.1. PCR-based microfluidics in molecular diagnostics

Analysis of nucleic acids through amplification by Polymerase Chain Reaction (PCR) is a powerful tool in molecular biology with many critical applications. PCR is based on the *in-vitro* enzymatic exponential amplification of a target DNA sequence, through a thermal cycling process of purified template DNA in the presence of additional DNA polymerase enzyme, nucleobases and forward and reverse primers suspended in an aqueous buffer solution. It can also be applied to generate DNA complementary to a starting RNA sequence, referred to as reverse transcription (RT) PCR, which is particularly useful in studying gene expression of viral RNA [1]. Apart from diagnostic purposes, sequencing and genotyping through PCR is used to enhance understanding of various diseases, record gene-dependent drug sensitivity or identify individuals [2]. Miniature PCR microfluidic devices has been introduced in the 90's [3–7] and since then investigated as tools for rapid [8–12] and accurate, high-throughput [13] analysis of biological samples for nucleic acids on portable systems that integrate on-chip target detection [14]. Point-Of-Care (POC) molecular diagnostic systems utilize this technology to deliver high throughput personalized sample analysis for clinical diagnosis and monitoring of diseases [15, 16]. Several research groups demonstrated complete systems, integrating all steps for PCR-based diagnosis of various pathogens in crude samples for field-use [17–19]; to an extent, the potential of POC molecular diagnostic technology is now met in some commercial cartridge-based PCR assays with pre-printed reagents for diagnosis of common infectious diseases and for drug sensitivity tests performed by trained personnel [20, 21]. Although, wide commercialization of POC molecular diagnostics requires the integration of all steps on a single chip, with sample-in-answer-out functionality at a low-cost, providing substantial benefits for the end-user compared to alternatives [22, 23]. The current potential of higher throughput, more reliable performance and target multiplexing in simpler, "smarter" and truly portable systems, connected in a dedicated health surveillance network is far from being fulfilled [24]. Commercial lab-on-chip diagnostic technology is showing cumbersome development, with several studies reporting that the lack of a critical or "killer" application has meant that lab-on-a-chip technology has not developed further [25–27]. Nonetheless, the need for manufacturing expertise and high development costs, combined with the lack of communication of research needs and strict regulatory frameworks in healthcare manufacturing highly contribute to the observed slow development [28, 29]. Manufacturing constraints often discourage experimentation on novel designs and experimental approaches in microfluidics, revealing the gaps between proof-of-concept academic results and their translation to meaningful applications. Commercial quality microfluidics are manufactured at large volumes, mainly with micro-forming processes and medical grade thermoplastics, due to low unit costs. Such processes are highly expensive for prototyping, due to the requirement of forming tool manufacturing and therefore only validated designs justify investment in a tool [30]. Alternatively, soft lithography of polydimethylsiloxane (PDMS) [17, 31], clean-room based photolithographic patterning or etching of SU-8, glass or silicon [18, 32] and material removal processes such as mechanical and laser micromachining [33] of a wide range of microfluidic materials are typically preferred for research prototyping and low volume production.

### 1.2. Polymer additive manufacturing for microfluidics

Unlike conventional processes, additive manufacturing (AM) is more accessible to researchers, due to the simplicity of use and low setup and running costs, allowing researchers to get hands-on developing custom devices [34, 35]. Microfluidics research into prototyping with

AM is essentially carried out with material extrusion, material jetting and vat photopolymerization AM technologies [36]. Material extrusion or Fused Filament Fabrication (FFF) has been popular in microfluidics prototyping for its wide range of biocompatible printable thermoplastics and has been used for printing various bioanalytical microfluidic devices, in mostly explorative projects. Representative studies included poly(ethylene terephthalate) (PET) devices for nanoparticle suspensions preparation and biochemical sensing [37], portable polylactic acid (PLA) immuno-array devices for detection of multiple cancer biomarkers in serum [38], a configurable polypropylene micro-reactor for bioanalytical applications [39], polycaprolactone (PCL) nervous-system-on-chip devices for cell co-cultures that enable physiological studies [40], wax-based extrusion of vascular 3D structures for chaotic mixing studies [41] and purpose-built systems for printing biodegradable, cell-loaded hydrogels for producing tissue scaffolds of clinically relevant sizes [42]. The main limitations of FFF are lower print resolution (when compared to vat photopolymerization and material jetting) and rough surfaces due to the stair-stepping effect created by deposited filaments, which can often lead to sealing issues [43]. Material jetting or inkjet 3D printing is preferred for printing microfluidics for its multi-material printing capability that enables sacrificial [44] and functional [45] material patterning. Additionally, drop-on-demand printing technology offers the highest resolution among AM processes, other than 2-photon lithography [46]. In microfluidics context, the process is often used in conjunction with UV-curable polymers for printing microfluidic prototyping tools as soft lithography moulds [47], active microfluidic circuitry components [48, 49] and various lab-on-chip devices for cell and bacteria studies, often focused on material biocompatibility [50–52]. Recently Yin et al exhibited a inkjet 3D printed droplet-based PCR device with complex 2.5D channel pattern to detect micro RNAs in MCF-10A cells and MDA-MB-231, which can act as reliable biomarkers in various cancer types [53]. The device was printed in commercial photopolymer with sacrificial wax and post-processed for wax removal according to specifically developed protocols [54]. Dielectric ink jetting has produced thin film based 2D printed devices with embedded elements for low-cost diagnostics [55], while inkjet 3D printing of SU-8 monolithic channel devices using dissolvable PMMA supports and embedded functional elements has also been demonstrated [56]. Inkjet 3DP performance is typically dependent on specific system capabilities and the available material options, and open source systems can print custom material formulations with their fluidic properties tuned to specific ranges. Limitations of the process involve low print speeds and the high cost of flexible inkjet printing systems.

Stereolithography (SLA) is the first 3D printing technique invented and patented in 1984 by Chuck Hull, the co-founder of 3D Systems [57]. It utilizes light projection to achieve prints in photocurable resin formulations. The capability to print small unsupported overhangs results in less post-processing, smoother and more complex geometries. The simplicity of SLA systems has been translated into several commercial low-cost printers, with some enabling third-party resins and printing of research formulations. There are two key points to consider when developing medical microfluidics with SLA: channel manufacturability and material biocompatibility. Manufacturability of enclosed volumes in SLA is dependent on the effective drainage of uncured resin, which can block channels [58, 59]. The minimum size of a monolithic microchannel is determined by part orientation, resin viscosity, and the print resolution (determined by a laser beam diameter or pixel size in the case of Digital Light Processing) [60]. Due to that limitation, research on microfluidics with SLA often focuses on adopting intelligent design and printing practices for sealing microfluidic prints [61, 62], the development of computational models for determining minimum printable feature size [63], the use of low viscosity SLA resins [64] and printing of soft lithography tools [47]. Studies on SLA printing of microfluidics often concludes that with appropriate bonding and good quality control, SLA-

based 3D printing shows great promise in future prototyping and low volume processing techniques for a variety of lab-on-chip platforms [65–68]. Biocompatibility of photopolymer prints is an additional issue that needs to be specifically addressed on an application basis. Commercial photopolymer resins for AM are formulations of acrylate or epoxy derived monomers and oligomers and typically contain a photoinitiator, which initiates polymerization on exposure to UV or visible light [69]. Most resins additionally contain additives to improve functional properties and printability [70]. Printed resins, in their semi-polymerized form are usually aggressive towards biological samples such as cells and DNA, through a variety of interactions with reactive moieties as residual monomers on, or leaching from, the printed surface, including commercial USP Class VI biocompatible formulations [71–73]. The impact of such interactions on a biological process as PCR can be detrimental in high surface-to-volume ratio (SVR) microfluidic architectures. Appropriate post-processing of photopolymer prints with increased temperature and UV light [51], organic solvents and water baths [52] has been shown in multiple studies to improve the biocompatibility of prints by reducing residuals content in photopolymer parts [52, 74]. Reducing leaching from the interface and reactivity at the interface by further static or dynamic coating of surfaces has also been a popular solution for 3D printed devices. Treatment is possible by exploiting the inherent surface reactivity of the prints to perform coatings with various coupling agents, endowing customised surface properties. Silane coupling agents [75] or biocompatible compounds as poly (ethylene glycol) (PEG), bovine serum albumin BSA, poly (vinyl alcohol) (PVA) [62] and poly-l-lysine [74] have been reported to improve functionality of photopolymer parts for applications such as droplet generation [53, 75], cell and bacteria culture and nucleic acids amplification [62, 74, 76]. Commercial biocompatible resins have been also successfully tested in low-risk applications such as skin and internal tissue contact for hearing aids and dental applications respectively, and for short period bacterial or cell cultivation experiments [77, 78]. In addition novel cytocompatible and photosensitive monomer formulations such as polyethylene glycol diacrylate (PEG-DA), ethoxylated trimethy-lolpropane triacrylate, poly(propylene fumarate)/diethyl fumarate and fumaric acid monoethyl ester based resins, have also been used in applications as cell-based microfluidics for cancer studies [79] and tissue scaffold printing [80, 81], demonstrating excellent biocompatibility.

## 1.3. Research motivation

Several studies on microfluidic prototyping with SLA have individually addressed the manufacturability issues associated with realisation of enclosed micro-features, the need for process standardization and the particularly reactive nature of commercial resins that makes them inappropriate for biological applications. Aim of this work is to define the processing protocols for the production of chips for stationary PCR, and to clearly demonstrate functionality. The research focusses on the production of chips for stationary PCR using open-source Digital Light Processing (DLP) SLA and a high-performance third-party acrylate resin. The 3D printed chips were post-processed to reduce PCR inhibiting residuals content and subsequently hydrophobically modified with silane agents to limit molecular adsorption instances. Silane-based modification protocols were based on dip-coating methodologies and characterized with Scanning Electron Microscopy-Energy Dispersive X-ray Spectroscopy (SEM-EDS) and contact angle visualisations and analysis prior to implementation on PCR chips. To evaluate performance, an infrared (IR) mediated PCR thermocycler for printed chips was developed. The system was developed to address issues associated with the problematic thermal behaviour of 3D printed flow-through chips [82], as the radiation directly heated the PCR mixture, enabling the evaluation of the PCR inhibitory effect caused directly by material

interactions rather than poor temperature control. The functionalized chips were evaluated by performing on-chip PCR to amplify a 75 base pair target sequence from genomic DNA in a protocol used as endogenous control in qPCR. The short target allowed rapid extension and denaturation times and was selected as a first step for proof-of-concept experimentation. Amplification reactions were performed on glass and fluorosilane coated chips and amplicons were subsequently detected off-chip with agarose gel electrophoresis and fluorescence and compared to control reactions ran at identical conditions on a high-speed PCR thermocycler. To highlight the effectiveness of the silane modification in limiting molecular adsorption occurrences, a non-treated chip was tested under identical conditions.

## 2. Methods

### 2.1. Micro-chamber PCR chip architecture

The stationary PCR chip geometry consisted of three cylindrical chambers of 3 mm diameter and 400 μm depth, interconnected with a 300x400 μm cross-section microchannel. Fillet features of 1 mm radius at the intersection of the chamber and the interconnected microchannels were inserted to eliminate rough edges that promote air bubble formation. The total surface area of the microfluidic chamber design was 36 mm$^2$, resulting in a total surface-to-volume ratio of approximately 5, and a 15 μL reaction volume with each interconnected chamber accommodating approximately 5 μL. Circular inlets of 400 μm nominal diameter were designed to fit and seal with a 200 μL pipette tip used for PCR mixture insertion. Temperature sensing was performed only in the central PCR chamber, through a 300 μm hole on the central chamber, designed for tight fitting a micro-probe thermocouple of the same diameter.

### 2.2. Characterisation of printed and post-processed resin interference with PCR

To simulate the behavior of a low SVR PCR fluidic device printed in SLA, low-volume printed specimens were exposed to PCR mixture during the reaction. The specimens were post-processed at different levels. Prior to PCR sample preparation, nine cubes (1 mm$^3$) were printed in Formlabs Hi Temp resin on the Autodesk Ember DLP SLA. The utilized resin is commercially developed for high-definition prints of high thermal stability and minimum deformation and its main application is 3D printing of thermoforming tools. Layer exposure time was selected at $2 \pm 0.1$ seconds, so that optimal feature definition is achieved, as it was visually observed (Figs 1 and 6). After printing, the specimens were washed in isopropanol, dried with air and post-cured in a UV oven (0.08 W/cm$^2$, 355 nm) for 1 hour, according to the manufacturer instructions [83] to achieve heat deflection temperature higher than 200˚C. At this stage, three cubes were placed in individual PCR tubes and sealed. The remaining six cubes were boiled and stirred in 100 mL deionized water. Three more cubes were then placed in PCR tubes and sealed. The final three cubes were boiled in deionized water for 1 hour, while placed under in the UV oven for 1 additional hour. They were then also stored in polypropylene tubes and sealed. A conventional PCR protocol to amplify a 1,000 base pair long target sequence was used for characterization. PCR mixture for nine 50 μL reactions (three replicates for each of the three post-processing conditions), plus three controls (positive, negative-no template, negative-no enzyme) was prepared. Each individual reaction contained 50% v/v Fail-Safe® PCR buffer (Premix D, Lucigen) with 4 mM MgCl$_2$ and 400 μM of each dNTP (10 μM), 5% v/v of each primer (SeCopAP1 forward TGTCAATGAATTGATGACCAATCATAAAGGA GTTTTTACTTGTAGGCTGGAGCTGCTTCG 10 μM and SeCopAP2 reverse CCGCCTTTAAGC AACTCGAATTATTTTGGGTATAGACTTTCATATGAATATCCTCCTTAG 10 μM), 1% v/v *Taq*

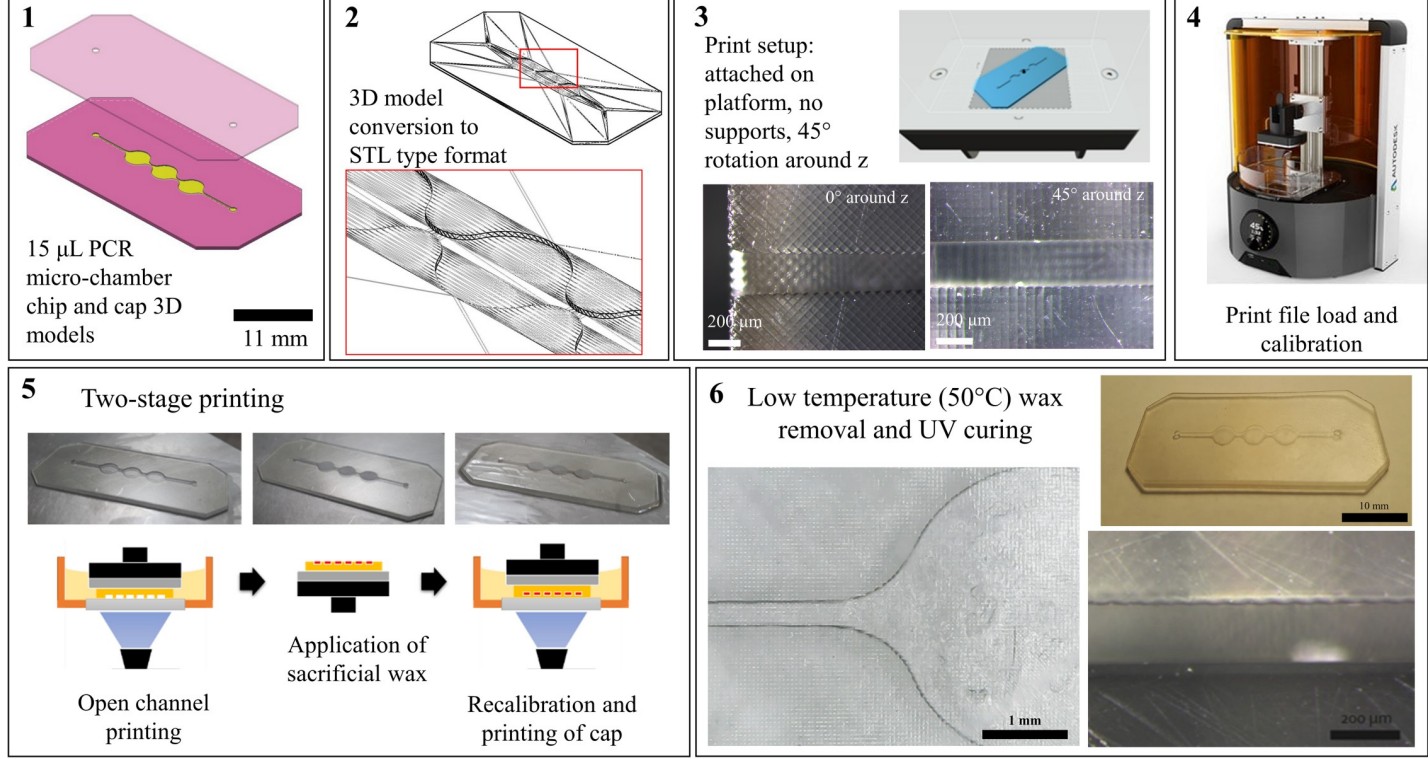

**Fig 1. Two-stage printing protocol schematic for printing monolithic microfluidic cartridges with DLP® SLA.** An open channel chip and a cap CAD model are developed and converted into STL type format. In AM software environment, the open-channel part is printed virtually positioned onto the printing platform and without support structures. The chip is then cleaned, and the channel cavities are filled with low melting point and viscosity sacrificial wax. The whole process takes place, while the chip is held on the platform by vacuum built during printing of the first layer. A cap part is then printed on the existing chip creating a monolithic channel. The chip is then removed from the platform, cleaned from excessive resin and dried. It is then heated at ~60˚ and the wax is flushed out with isopropanol and hot water. The printed chip can then be further post-processed and treated.

DNA polymerase and 1% v/v plasmid DNA template solution (30 ng/µL) in molecular biology grade water. The reaction mixture was pipetted into the nine tubes containing cubes, and into three empty tubes as controls. All twelve tubes were gently mixed, pulse spun and then run in a BioRad C1000 PCR thermocycler. The thermocycling protocol included an initial denaturation step at 95˚C for 5 minutes, followed by denaturation at 95˚C for 30 seconds, annealing at 52˚C for 30 seconds and extension at 68˚C for 60 seconds repeated for 35 cycles with a final extension step of 5 minutes. After thermocycling, 5.5 µL of 10X loading dye was added to each PCR tube. The contents of each tube were mixed and 10 µL aliquots of each sample were loaded with 10 µL of DNA ladder (80–10,000 base pair) and resolved in a 1% (w/v) agarose gel for 35 minutes at 90V and imaged on a BioRad transilluminator.

## 2.3. Chips fabrication with DLP- stereolithography

A two-stage printing process based on DLP-SLA and the Formlabs Hi Temp resin was developed for printing monolithic micro- channel and micro-well features based on the use of sacrificial, low-viscosity paraffin wax. An open channel chip with a 2.5D channel pattern was first printed, parallel to the projection plane of printer and attached on the platform without support structures. The platform was then removed with the print, the open channel chip cleaned with isopropanol and dried before applying sacrificial paraffin wax (Paraffin wax, melting point: 53–57˚C (ASTM D 87), Sigma Aldrich). After wax deposition, a warm glass slide is

pressed towards the deposited wax to improve surface quality of the subsequent printed layer. The entire cleaning and wax deposition processes occur with the printed part held on the platform by the vacuum generated during printing of the first layer. The platform is then placed back on the printer and the cap is printed on the existing part, sealing the channels. After a chip was printed, it was removed from the platform, washed in isopropanol and dried. It was then heated to 50–60˚C to melt the wax, which was flushed out with hot isopropanol and subsequent water washes. A schematic of the two-stage printing process is shown in (Fig 1). The chips were then UV cured for one hour in a purpose-built 355 nm UV oven at 0.08 mW/cm$^2$. Printed and post-UV cured chips were further washed with boiling hot water internally for one hour. Following the functionalization of the chips described in the following paragraph, the chips were fitted with a K type thermocouple with a probe of 300 μm diameter and sealed with UV curable adhesive.

## 2.4. Hydrophobic functionalization of SLA prints

To internally treat the micro-chamber chips printed with SLA, a perfusion coating methodology was considered most appropriate. Glass and fluorosilane hydrophobic functionalization protocols were initially developed as dip coating methodologies to functionalize 2D printed and post-processed disks of 10 mm diameter and 1 mm thickness. Those specimens were used for the evaluation of coatings with contact angle and SEM-EDS prior to implementation on 3D printed chips.

**2.4.1. Dip coating of 2D SLA prints.** To functionalize the 2D prints with fluorosilane, after post-processing as described in 2.2, the prints were dipped into reactive siloxane solution (G790, Wacker) and left for 30 minutes before being removed and dried at room temperature. A 95:5 ethanol: deionised water solution was adjusted to have a pH of 4.5 using glacial acetic acid. The fluorosilane coupling agent (1h,1h,2h,2h-perfluorooctyltriethoxysilane, Sigma Aldrich) was then added to the ethanol: deionised water solution yielding a 2% v/v concentration. The solution was stirred for 5 to 10 minutes to instigate silanol formation. The siloxane pre-treated 2D prints were then moved into the fluorosilane solution and stirred for 3–4 hours at room temperature. After functionalization the specimens were washed with deionised water, dried and stored in petri dishes prior to evaluation. The siloxane pre-treatment and the subsequent fluorosilane coating mechanism s are illustrated in (Fig 2).

The glass hydrophobic coating involved similarly dipping of the 2D post-processed prints in reactive siloxane agent for 30 minutes, before removed and dried at room temperature. The prints were then dipped into Sigmacote® and agitated for few seconds. They were then cured at 130˚C for 15 minutes and then cooled to room temperature over a period of 6–8 hours toa void thermal shocks and therefore cracking and flaking on the coated layer. The siloxane pre-treatment and subsequent glass coating mechanisms are illustrated in (Fig 3).

**2.4.2. Perfusion coating of SLA printed monolithic micro-channels.** To functionalize the chips with fluorosilane, post-processed and dried chips had the micro-chamber filled with the reactive siloxane solution, which was left for 30 minutes, then flushed away with pressurized air. The chip was then dried. The fluorosilane coupling agent was added to a pH adjusted ethanol: deionised water solution prepared as described for the 2D prints. The dried siloxane pre-treated chips were connected to the chip bracket designed for the IR system as described in 2.4. M6 fitting nuts (Omnilok, Kinesis), silicone ferrules and 1/16" diameter tubing were used to connect the chip to a Harvard PHD 22/2000 syringe pump. The fluorosilane solution was then cycled through the chip for 2 hours at flow rates of 0.1–1 ml/min. During this period, the chip was heated to approximately 80˚C. The functionalized chips were disconnected from the manifold and washed thoroughly with PCR grade water and dried. Similarly, to

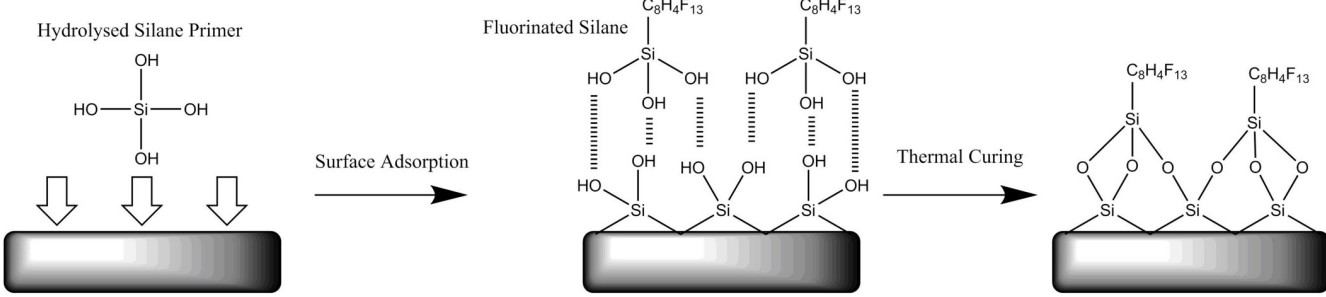

**Fig 2. Schematic representation of the consecutive treatment with siloxane and fluorosilane agents to functionalize a 3D printed acrylate surface with fluorosilane.** The siloxane treated acrylate surface is dipped into hydrolysed fluorosilane agent. Hydrogen bonds are initially formed between the primer and the coating agent, which are covalently bonded to each other after thermal treatment.

functionalize the chips with glass, the post-processed chips were filled with the reactive siloxane solution, left for 30 minutes and flushed and dried with pressurized air. Sigmacote® was then manually inserted and cycled in the chamber with a mechanical pipette for a few seconds before the excessive solution was flushed out. The chips were then cured at 130°C for 15 minutes and cooled to room temperature over a period of 6–8 hours. Functionalized chips were fitted with a thermocouple as described in 2.2.

## 2.5. Infrared thermocycler development

The IR thermocycler employed two active elements for heating and cooling and a chip bracket. The heating element was a 12 W, 850 nm PCB mounted IR LED cluster (RS Components), consisting of nine individual LEDs in a 3 X 3 square configuration of 25x25 mm, bonded with electrical insulating and heat conductive compound on an anodized aluminium heatsink. Each LED provided a radiant flux of 1.3W under its highest permissible operation, and half of its intensity at a 45° viewing angle. Considering the spacing between the centre points of individual LEDs, the chip was located at an effective height of 2.5 mm above the LED top surface to achieve homogeneous IR emission and even heating of the PCR mixture. The cooling element was a 24V DC centrifugal air fan (RS Components), fitted on the side of the heatsink/ LED array and chip bracket configuration (Fig 4). Both active elements were driven by MOS-FET transistors and controlled by an 8-bit Arduino-based microcontroller. PID (proportional-

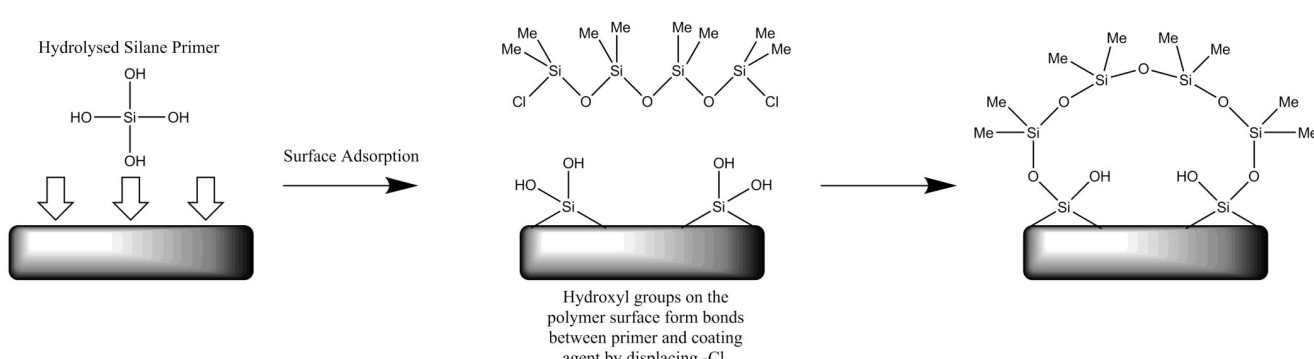

**Fig 3. Schematic representation of the consecutive treatments with siloxane and glass agents to functionalize a 3D printed acrylate surface with glass.** The siloxane treated surface is dipped into Sigmacote® glass in heptane solution, and bonds are formed between the primer and the glass agent by displacing -CL.

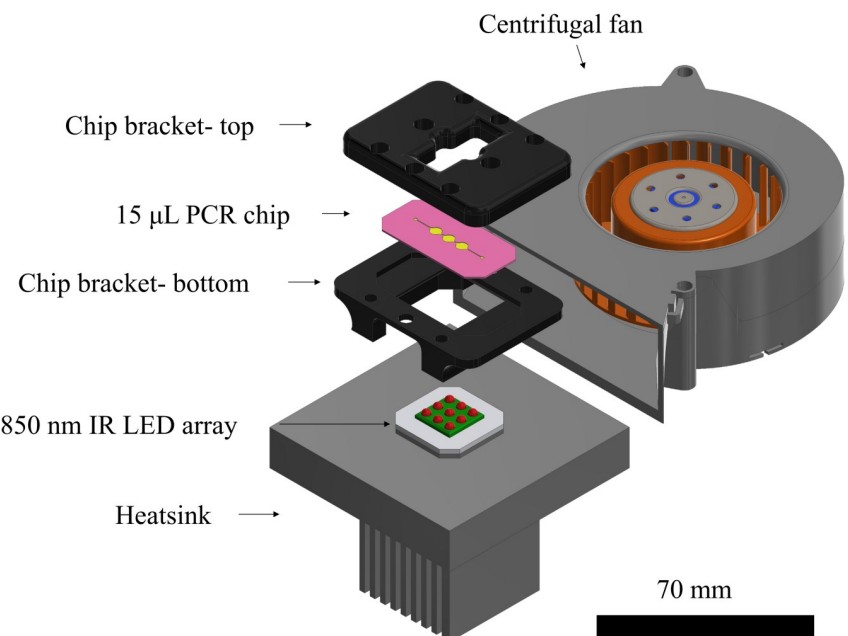

**Fig 4. Schematic of an infrared (IR) PCR thermocycler for a 3D printed microchamber PCR chip.** CAD assembly of chip bracket, chip, IR LED cluster and heatsink and a centrifugal fan. The sides of the bracket exposed the bottom chip surface allowing cooling of both the chip and the LED cluster by the fan and insulating the area around the temperature sensor. Copper tape was used to cover internal surfaces of the bracket, reflecting IR towards the chip, and improving the heating rate. The system provided excellent repeatability in thermocycling for PCR, enabling the characterisation of the PCR inhibition exclusively owed to the performance of the printed polymer, eliminating inhibition owed to temperature control. The developed thermocycling platform and its modular character based on open-source electronics additionally provides a basis further exploration of resin-3D printed microfluidic geometries for Point-of-Care analysis of nucleic acids.

integral-derivative) control was implemented into the Arduino code and the gains were calculated for the 2-step PCR protocol described in 2.5 to achieve minimal temperature overshoot and increased stability. Temperature sensing was performed with the thermocouples embedded in the PCR chamber of the 3D printed chips during fabrication. A chip bracket was designed, and 3D printed in Autodesk Ember DLP SLA with PR48 black prototyping resin. The bracket design served a series of purposes, including housing and sealing of the chip during PCR, interfacing commercial tubing for post-processing and perfusion treatments, improving air-cooling, reflecting IR toward chip chamber and locating the chip at the effective height for optimum IR transmission. M6 holes were tapped on the 3D printed bracket for attaching Omnilok M6 microfluidic fitting nuts (Kinesis, UK) and 1/16" microfluidic tubing, used during post-processing and coating treatments. Regular plastic M6 fitting nuts and pressure sensitive tape were used for sealing during PCR.

## 2.6. Two-step PCR protocol for on-chip reactions

To demonstrate PCR functionality from 3D printed micro-well devices, genomic DNA template from a DNA pool was used to amplify a 75 bp target sequence using CD8a quantitative PCR primers based on a two-step PCR thermocycling protocol. PCR concentrations were optimized to take into account molecular adsorption. Phire II hot start polymerase was specifically selected for its extension rate of 10 s/kb, which allows shorter extension times even in environments that inhibit enzyme activity and compared to reactions performed with *Taq* polymerase. The protocol was performed on 3D printed chips treated with fluorosilane and Sigmacote®. A

non-treated chip was also tested for control purposes. To establish reliable controls for the IR system and for the reaction itself positive and negative control reactions were performed on a Philisa high-speed thermocycler under an identical PCR protocol and similar heating rates. Individual reaction volumes for IR PCR (15 μL) and Philisa (25 μL) positive control reactions contained 20% Phire buffer (5x), 200 μM dNTPs, 2μM each of the CD8a forward and reverse primer respectively (CD8a Forward `CACCTGGAAGCGACTTAGATACTGT`/`CD8A Reverse CCCTACTCAGCCTCGCGTTA`), 10% (v/v) Phire II polymerase and genomic template (4000 copies per reaction) in molecular biology grade water. The 15 μL PCR mixtures were pipetted into the micro-chamber with a mechanical pipette and a 200 μL tip. The PCR thermocycling protocol included an initial denaturation step at 95°C for 60 seconds, followed by 50 cycles of denaturation at 95°C for 2 seconds and an extension- annealing step at 68°C for 5 seconds, followed by a final extension step at 68°C for 60 seconds. This protocol was implemented for both IR and Philisa control reactions. After thermocycling, amplicons were loaded with 6x Purple loading buffer and 5μL aliquots were loaded and resolved on a 2% (w/v) agarose in gel in TAE buffer at 90V for 45 minutes along with a low molecular weight DNA marker (NEB) of 25–766 bp using 0.5 and 1 μL loadings. For demonstration purposes, the faint amplicons retrieved from non-treated chips ran on agarose gel for approximately 20 minutes, before fully absorbed in the agarose gel after a few additional minutes. Gels were imaged in a BioRad trans-illuminator system and bands were analysed using GelAnalyzer. Target amplicon concentrations were calculated based on Beer-Lambert law and using DNA ladder bands as standards.

## 3. Results and discussion

### 3.1. Characterisation of printed and post processed resin interference with PCR

The reactions performed in the presence of low volumes of printed material, indicated that basic post-processing with only isopropanol washes and UV light did not efficiently reduce PCR inhibiting residuals, so that a target concentration consistent with the control reaction is yielded (Fig 5(B): bands 2–4). In contrast, the reactions performed with printed specimens washed with hot water and further UV-cured (Fig 5(B): bands 5–9) yield amplicon concentrations consistent with the positive control [Fig 5(B): band 11]. Negative control reactions lacking template DNA (Fig 5(B): band 12) and enzyme (Fig 5(B): band 13) did not generate any product as it was anticipated. Those results (S1A in S1 File, Fig 5) address PCR inhibition, due to leachates, but also due to biomolecular adsorption, which in this case has minimum impact due to the very low SVR of the cube specimen (0.12). The results demonstrate a strong PCR inhibiting effect of the SLA polymer when this is post-processed minimally with isopropanol and UV light, while hot water baths are shown to reduce inhibiting residuals. UV curing reduced PCR inhibiting residuals by inducing their consumption into the polymerized material and increasing the degree of cross-linking. Water and isopropanol further dissolve print residuals to remove them from the printed surface. Therefore, the printed polymer is compatible with PCR for fluidic and potentially milli-fluidic chamber-based devices when post-processed appropriately. However, the evidence is partially representative for higher SVR architectures, which are anticipated to enhance the inhibiting effect of biomolecular adsorption in on-chip reactions.

### 3.2. Chip design and printing

The three-chamber architecture was observed to be more rigid during printing, with easier application of sacrificial material and fitting of thermocouples compared to a larger single-chamber design. Moreover, the geometry repeatedly induced no bubble formation during PCR

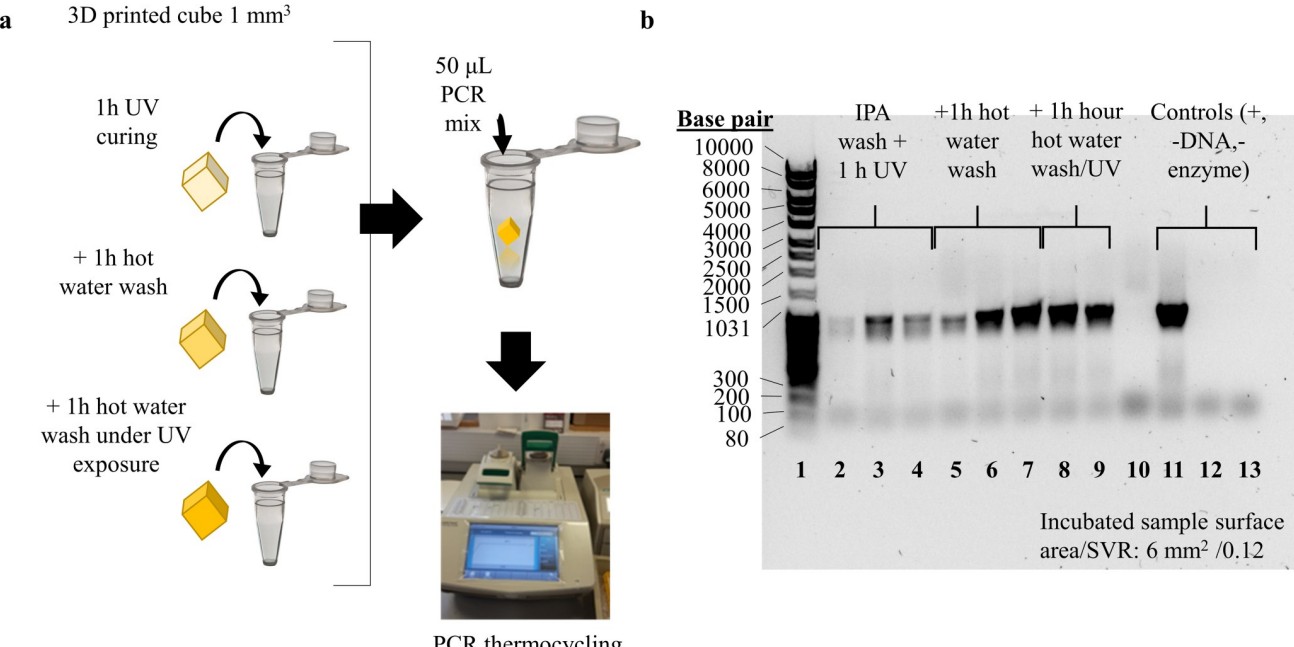

**Fig 5. Mock PCR experiments for characterisation of printed resin interference with PCR. (a)** Schematic illustration of mock PCR experiments for low SVR incubated prints. Printed samples representative of consecutive post-processing steps including isopropanol (IPA) wash, UV curing and hot water baths were incubated into the PCR mixture and remained until the completion of the reaction, **(b)** Gel electrophoresis imaging of separated PCR amplicons from mock reactions: low SVR printed samples incubated in PCR mixture during PCR. Samples post-processed with high intensity UV light and hot water washes (bands 5–9) showed minimum interaction with DNA amplification, unlike the ones post-processed only with isopropanol washes and/or UV curing (bands 2–4) (S1A_raw_images in S1 File (DOI: 10.25405/data.ncl.12320501)).

mixture insertion. Printing time of an open channel chip and the cap was 3 and <1 minute respectively using 50 μm printed layers and considering total nominal chip thickness of 0.9 mm. Total printing time for a chip was 10 minutes, including wax deposition and removal. In this work, one chip was printed per print run. However, multiple chips can fit in the platform of most DLP SLA systems to increase productivity by maintaining identical printing times irrelevant to print surface area. The total post-processing time including simultaneous UV curing and water washes for a printed cartridge was approximately 1 hour, with the capability to post-process multiple chips simultaneously depending on the setup. By following the manufacturer instruction for post-UV with an 80 mW/cm$^2$ light source, no distortion was observed at a level that would impair the performance of the printed device. The chips were designed to accommodate 15 μL fluid volumes, which were precisely loaded using a mechanical pipette. No flatness issues were observed, while the small chip area and thickness allowed for a level of flexibility. The complexity of channels achievable with the proposed methods is based on the melt viscosity of the wax, which determines the backpressure required for wax removal and the feasibility depends on the bonding strength between the chip and the cap. The low melt viscosity of the paraffin wax, in combination with the unibody microchannel formed can deliver more complex 2.5D channels, such as heat exchanging architectures found in flow-through thermocycling setups.

### 3.3. Hydrophobic functionalization of SLA printed surfaces

The contact angle of fluorosilane and glass coated samples was approximately 115˚ (Fig 6B, 6E and 6H) and 135˚ respectively (Fig 6C, 6F and 6I). For a non-treated surface, the value was 80˚

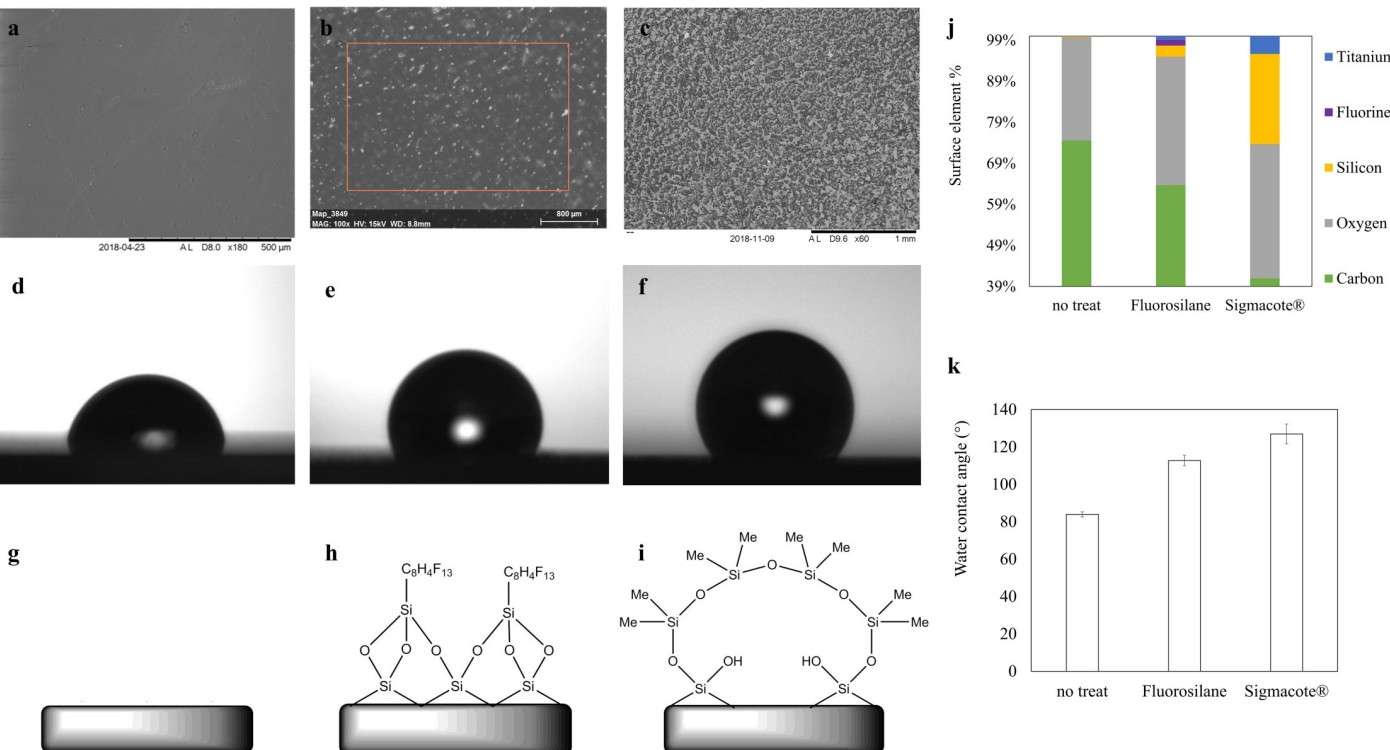

**Fig 6. Characterisation of silane-functionalized 3D-printed disk specimens.** SEM images, contact angle visualizations and surface chemistry schematics of **(a, d, g)** a non-treated, **(b, e, h)** a fluorosilane and **(c, f, i)** a glass modified printed 2D surface, with wetting contact angle values of **(d)** 80˚, **(e)** 115˚ and **(f)** 135˚ for each respective condition for a 0,05 μL deionised water droplet, **(j)** SEM-EDS surface elemental composition estimation for two silane treatment conditions, compared to a non-treated printed surface. (k) contact angle results summary. (S2A_SEM_contact_angle in S1 File (DOI: 10.25405/data.ncl.12320837), S2B_contact_angle in S1 File (DOI: 10.25405/data.ncl.12320849), S3_Fig_6_SEM_EDS in S1 File (DOI: 10.25405/data.ncl.12320861)).

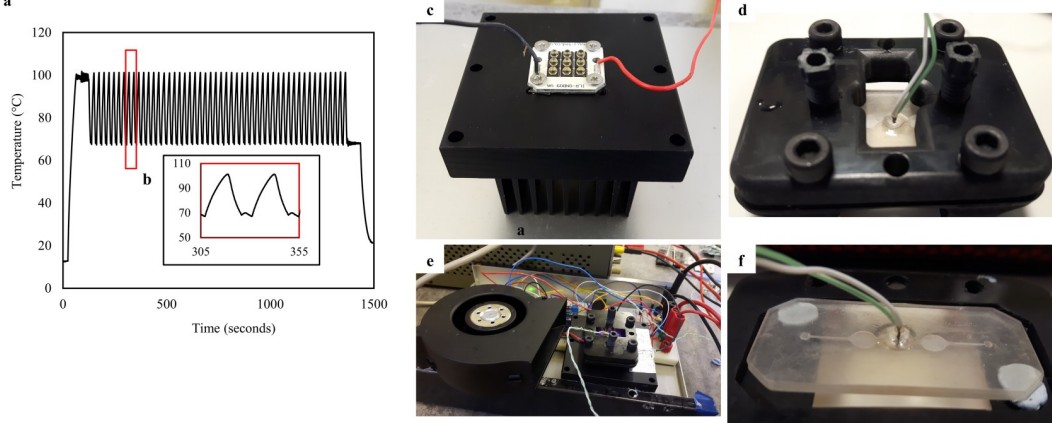

**Fig 7. IR PCR thermocycling performance and system configuration. (a)** A two-step PCR thermocycling profile for 50 cycles and one-minute initial denaturation and final extension steps for 15 μL on-chip reactions. The temperature profile was recorded on a thermocouple embedded 3D printed PCR chip in real-time during PCR thermocycling on a fluorosilane modified chip, **(b)** zoom-in one two-step PCR cycle temperature profile, **(c)** a 3D printed chip modified with fluorosilane and fitted with an embedded thermocouple, mounted on the bottom chip bracket prior to PCR, **(d)** A 3D printed and post-processed, non-treated PCR chip, **(e)** the IR thermocycler, **(f)** PCR chip assembled on the IR chip bracket prior to reaction. (S4_2_step_PCR-temperature_profile in S1 File (DOI: 10.25405/data.ncl.12320864)).

(Fig 6A, 6D and 6G) (S2A, S2B in S1 File). Elemental composition analysis with SEM EDS showed distinctive elemental signals corresponding to silicon and fluorine and increased presence of oxygen when compared to non-functionalized samples, which is indicative of silane chemistry (Fig 6J) (S3 in S1 File).

The perfusion coating steps lasted 2 and 1 hour for the fluorosilane and glass layers respectively. Considering a 10-minute print time per chip and one hour of post-processing (if UV curing and water washes are combined), a minimum of 2–3 hours is required to fabricate one or more fully post-processed and silane-functionalized PCR micro-chamber chips. The number of chips that can be produced in this period depends on the capabilities of the post-processing and treatment setup. As far as the silane coatings are concerned, it is recommended that surface-modified chips are stored overnight prior to use to allow for complete curing of the coatings.

### 3.4. Thermocycling performance

The fluid volume of 15 μL within the chips was required to undergo a two-step PCR protocol with the temperature oscillating between 95°C and 68°C for denaturation and extension-annealing steps respectively. The thermocycling performance was experimentally determined during PCR reactions using the thermocouple embedded in the printed chips during fabrication as described in 2.2 (Fig 7D and 7F). Total thermocycling time for the IR and control reactions were 25 and 20 minutes, respectively, with the experiments performed at room temperature of approximately 20°C. The average heating rate of the IR system was 4.1°C/s, with a maximum value of 14.3°C/s. The average and maximum cooling rates were -5.1°C/s and -12.5°C/s, respectively (Fig 7A and 7B) (S4 in S1 File). The heating rates and cycle times were identical for untreated, fluorosilane treated, and glass treated chips. The PID code was optimized for 15 μL reaction mixtures and achieved thermal stability of ±1.5°C during initial overshooting, and less than ±0.25°C in the steady state. The stabilisation time was 5–10 seconds. In terms of the material performance in thermocycling, the reaction chamber withstood the pressure built during PCR and the material remained rigid with no visual signs of deterioration. The system design with radiation heating and stationary thermocycling enables testing of more complex PCR geometries, which can serve additional purposes such as DNA extraction, reagents selective delivery and optical detection. Functionalities such as fluidic control and optical detection can be incorporated in a modular fashion exploiting the open-source electronics platform. Finally, the compact system design allows for portability, which is desirable in POC applications.

### 3.5. On-chip PCR performance

By examining the agarose gel bands and their respective intensity profiles, a highly specific 75 bp target concentration is observed for the reactions performed on the IR system and 3D printed fluorosilane (Fig 8(A): Lane 5) and glass (Fig 8(B): Lane 4) modified chips. PCR products of those reactions were of identical sequence length and slightly lower intensity to the positive control reaction respective bands in (Fig 8(A): Lane 3) and (Fig 8(B): Lane 2). Negative control reactions, lacking template DNA performed on printed chips, resulted in no amplification product (Fig 8(A): Lane 4, 8(b): Lane 5), similarly to the negative control reactions ran on the Philisa thermocycler (Fig 8(A): Lane 2, 8(b): Lane 3). The PCR performance was consistent with the controls for both glass and fluorosilane coatings. Slightly lower concentrations of amplicon were observed from the functionalized 3D printed chips compared to the control reactions which apart from surface reactivity are potentially influenced by altered temperature stability and thermal transition rates, both of which can influence enzyme activity. Non-treated

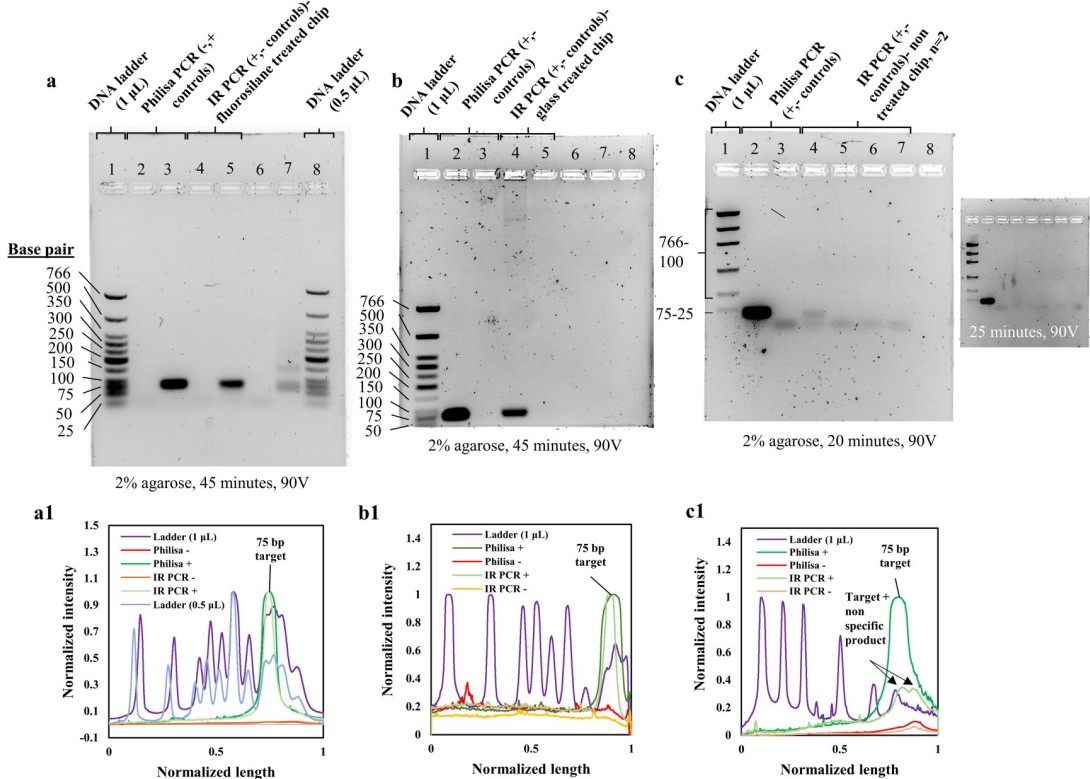

**Fig 8. On-chip nucleic acid amplification reactions on 3D-printed and silane-functionalized micro-chamber chips. (a)** Band intensity analysis of positive and negative (no DNA template) control reactions on Philisa (Lane 2,3) and IR PCR thermocycler on fluorinated 3D printed chips (Lane 4,5,6,7). Loading of 1 (Lane 1) and 0.5 μL (Lane 8) low molecular weight DNA marker allowed accurate quantification of amplicon based on band-intensity. **(a1)** Agarose gel intensity profile plots generated in ImageJ. Base pair lengths of DNA marker have been tagged for reference to target specificity. **(b)** Band intensity analysis of positive and negative (no DNA template) control reactions on Philisa (Lane 2,3) and IR PCR thermocycler on glass-coated 3D printed chips (Lane 4,5), run with 1 μL (Lane 1) loading of low molecular weight DNA marker. **(b1)** Agarose gel intensity profile plots generated in ImageJ. PCR product bands for positive and IR PCR reaction representative peaks were tagged. **(c)** Band intensity analysis of positive and negative (no DNA template) control reactions on Philisa (Lane 2, 3) and IR PCR thermocycler on non-treated 3D printed chips (Lane 4,5,6,7), run with 1 μL (Lane 1) loading of low molecular weight DNA marker. The gel run for approximately 20 minutes less than the time for the gels with amplicons from treated chips, to visualize the faint amplicon, as the latter was fully absorbed in longer gel electrophoresis runs (inset in c). As a result, the ladder in this gel image is not fully separated. **(c1)** Agarose gel intensity profile plots generated in ImageJ. PCR product and primer bands for positive and IR PCR reaction representative peaks were tagged. (S1A_raw_images in S1 File (DOI: 10.25405/data.ncl.12320501), S1B_Amplicon_band_intensities_plots in S1 File (DOI: 10.25405/data.ncl.12320396)).

printed chips explicitly produced very faint or no PCR product (Fig 8(C): Lane 4,6). The gel containing the PCR amplicon retrieved from the non-treated chip did not run for the full period of time as the previous gels. The gel electrophoresis was stopped at 20 minutes and the gel was imaged to visualize the faint amplicon, which was later absorbed after few minutes of additional electrophoresis of the same gel (Fig 8, (c inset)) (S1A in S1 File). This resulted in partially separated DNA ladder bands.

Enzyme adsorption rather than DNA template is here considered the main source of PCR inhibition, which is in agreement with similar literature [62, 71, 74]. Enzyme can be anchored on the printed surface and become limiting during early PCR cycles, resulting in low target concentrations. In addition, PCR compounds such as denatured DNA, primers, dNTPs and Mg ions are further prone to adsorption and might also become limiting, due to surface reactivity, further inhibiting the reaction. Silane treatments exploit the surface reactivity to create a

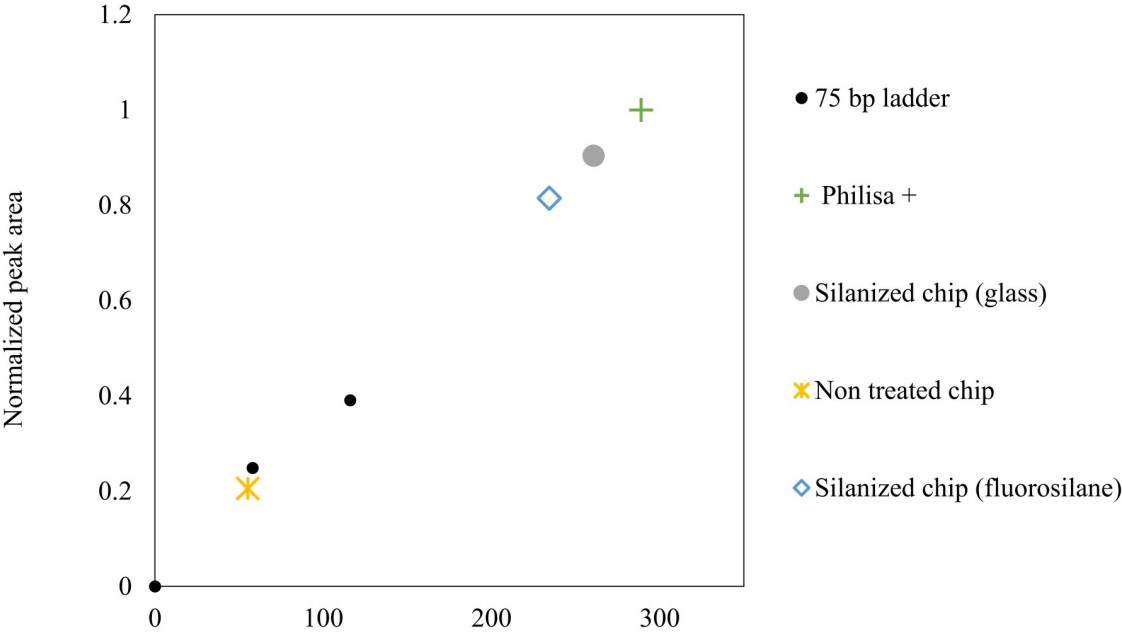

**Fig 9. Band intensity-based estimation of amplicon concentration for on-chip reactions.** Calculations of concentration were based on calibration bands obtained from low molecular weight (25-766bp) DNA ladder of 0.5 and 1 μL loadings, resolved in 2% agarose (45 min, 90V) (S1B_Fig_8_Amplicon_band_intensity_plots in S1 File (DOI: 10.25405/data.ncl.12320396)).

hydrophobic passivated surface, which significantly limits the adsorption interface and level of reactivity. PCR components could not be adsorbed on the hydrophobic surface and the reaction was facilitated. Considering similar literature results, enzyme is the main adsorbed PCR component [84, 85]. The proposed hydrophobic treatments are appropriate for reducing biological adsorption in 3D printed microfluidics. The estimation of target concentrations on on-chip reactions was performed on GelAnalyzer by analyzing the peak areas of the 75 base pair long fragment bands for the two loadings of 1 and 0.5 μL shown in (Fig 8A and 8a1: Lane 1 and 8) and constructing a calibration curve (S1B in S1 File). The curve was then used to calculate the concentration for the bands of the control and on-chip reactions (Fig 9) (S1B in S1 File). The calculated concentration values were 288 ng/ μL for the positive control reaction, 260 ng/ul for the glass modified chip and 234 ng/ μL for the fluorosilane modified chip. The respective value for the non-treated chip was calculated at 55 ng/ μL. Such low concentration levels compared to the controls can argue that denatured, short 75 base pair target sequences might also be adsorbed into the channel walls. If it is assumed that the efficiency of the control reaction is 100%, then the ratio of concentration of an on-chip reaction to the concentration of the control would indicate the PCR efficiency. This was calculated at 81% and 90% for the fluorosilane and glass modified chips respectively. The same value for a non-treated chip was 20%.

The capability for customising the surface of SLA prints for reducing non-specific adsorption during PCR, means that further experimentation is possible to create functional devices for nucleic acid amplification applications that can be printed and functionalized in-house at very low costs. This characteristic is highly attractive for prototyping POC molecular diagnostic platforms, as well as cell-based lab-on-chip, where the behaviour of the surfaces that interact

with biological fluids is of utmost importance. The proposed methods can form the first stage towards optimization and standardization.

## 4. Conclusion

Low-cost research prototyping of customised PCR microfluidics is highly desirable within the molecular diagnostics community, driven by the increased demand for low-cost POC molecular diagnostics. The latter is almost exclusively based on amplification of nucleic acids. In this paper we have presented specific processing routes to repeatedly deliver low-cost microfluidic chamber PCR devices printed with DLP SLA in a two-step printing protocol with sacrificial wax. The concept of sacrificial wax is known for realising various enclosed microstructures including microfluidic channels [86] and has been recently reported for fabricating complex flow-through PCR architectures with inkjet 3D printing [53]. In this work, the concept was extended to DLP SLA. The presented method of printing utilizes low viscosity paraffin wax to deliver a unibody non post-processed microfluidic chip in less than 10 minutes. The water-like viscosity of the wax enables complex channel patterns and sizes of few hundred microns. The minimum achievable dimension for enclosed channels is mostly appropriate for diagnostic applications, where the fluid samples range typically at the micro-litre scale for clinical relevance. The raw material unit cost of the printed chips was below £0.20, not including a temperature sensing element or the cost of the functionalization reagents. Indicatively, the cost of the temperature sensor was £0.55 and the functionalization agents £0.6 and £0.02 per chip for fluorosilane and glass respectively. The suggested protocols pose an attractive alternative for microfluidics prototyping, as it requires no tool manufacturing, thus enabling developers to promote or discard ideas faster and cheaper than with conventional processes. In addition, the study showcases that reactive resin chemistries may provide a basis for developing surface treatments to achieve on-demand material performance and hence enable challenging biological applications, such as PCR. The process requires no clean-room setup, but a print post-processing station which can be fitted in any basic research environment. The mask-less patterning capability of DLP SLA together with flexible resin chemistries mean that further experimentation is possible. However, extending the work beyond prototyping volumes is largely a commercial decision, based on the price per chip. Future research will be focused on the optimization of the microfluidic design, material, instrumentation and biology for this method of chip fabrication and temperature control, and the incorporation of further functionalities towards fully 3D printed molecular diagnostics. Additional interest emerges from the potential to couple DLP SLA with CNC micromachining and functional ink-jetting in a hybrid system to automatically produce functional micro-assemblies, including microfluidics in a single manufacturing environment.

## Supporting information

**S1 File.**
(RAR)

## Acknowledgments

The authors would like specifically to thank Dr Stuart Oram, as well as the entire QuantuMDx assay team, for their support and cooperation.

## Author Contributions

**Conceptualization:** Charalampos Tzivelekis, Kenny Dalgarno.

**Data curation:** Charalampos Tzivelekis.

**Formal analysis:** Charalampos Tzivelekis.

**Funding acquisition:** Kenny Dalgarno.

**Investigation:** Charalampos Tzivelekis.

**Methodology:** Charalampos Tzivelekis, Pavlos Sgardelis.

**Project administration:** Charalampos Tzivelekis.

**Resources:** Kevin Waldron.

**Software:** Charalampos Tzivelekis, Pavlos Sgardelis.

**Supervision:** Kevin Waldron, Richard Whalley, Dehong Huo, Kenny Dalgarno.

**Validation:** Charalampos Tzivelekis.

**Visualization:** Charalampos Tzivelekis.

**Writing – original draft:** Charalampos Tzivelekis.

**Writing – review & editing:** Charalampos Tzivelekis, Kenny Dalgarno.

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
