## [Decision Letter · Decision Letter 0]

6 May 2020

PONE-D-20-09489

Fabrication routes via projection stereolithography for 3D-printing of microfluidics for nucleic acid amplification

PLOS ONE

Dear Dr Tzivelekis,

Thank you for submitting your manuscript to PLOS ONE. After careful consideration, we feel that it has merit but does not fully meet PLOS ONE’s publication criteria as it currently stands. Therefore, we invite you to submit a revised version of the manuscript that addresses the points raised during the review process.

We would appreciate receiving your revised manuscript by Jun 20 2020 11:59PM. To enhance the reproducibility of your results, we recommend that if applicable you deposit your laboratory protocols in protocols.io, where a protocol can be assigned its own identifier (DOI) such that it can be cited independently in the future. For instructions see: http://journals.plos.org/plosone/s/submission-guidelines#loc-laboratory-protocols

We look forward to receiving your revised manuscript.

Kind regards,

Jonathan Claussen

Academic Editor

PLOS ONE

5. Please ensure that you refer to Figure 2-4 and 6 in your text as, if accepted, production will need this reference to link the reader to the figure.

Reviewers' comments:

Reviewer's Responses to Questions

**Comments to the Author**

1. Is the manuscript technically sound, and do the data support the conclusions?

Reviewer #1: Yes

Reviewer #2: Yes

2. Has the statistical analysis been performed appropriately and rigorously? 

Reviewer #1: N/A

Reviewer #2: Yes

3. Have the authors made all data underlying the findings in their manuscript fully available?

Reviewer #1: Yes

Reviewer #2: Yes

4. Is the manuscript presented in an intelligible fashion and written in standard English?

Reviewer #1: Yes

Reviewer #2: Yes

5. Review Comments to the Author

Reviewer #1: The manuscript by Tzivelekis and coworkers described the fabrication and testing of a 3D-printed PCR device and thermocycler. Digital Light Processing (DLP) stereolithography (SLA) was used as a high-res 3D printing technique for the fabrication of a monolithic micro-channel PCR device. A UV light and solvent bath was further performed to limit biomolecular adsorption. An IR mediated thermocycler was also built that is fully compatible with the 3D printed PCR device. This device was tested for amplifying a 75 bp subgenomic DNA sequence, the amplicon of which was compared with commercial PCR machines. The authors gave a very nice introduction summarizing the miniaturization of PCR machines using microfluidics and 3D printed devices for nucleic acid analyses. They also correctly identified existing challenges for miniaturizing PCR through 3D printing. The device and the thermocycler have been successfully verified by amplifying DNA sequence through PCR protocol. However, the current manuscript is too preliminary to be published. The authors shall expand more on the application end to validate the device and demonstrate the fluidic controllability.

1. The miniaturized PCR device was claimed to be a microfluidic device. However, there was very little fluidic control demonstrated. The device looks more like a microwell to the reviewer. The authors shall specify more on the fluidic control, e.g. how was the PCR reagents delivered? Ideally, a better device design with fluidic control shall be demonstrated.

2. A related question is the level of complexity that DLP SLA printing would allow. The two-stage dip coating of the hydrophobic layer on chip may increase the production period and difficulty.

3. The authors claimed that “the system is qPCR ready by integration with a fluorescent microscope and qPCR DNA stains. However, this was not demonstrated experimentally in the manuscript.

Reviewer #2: The goal of this work is to describe processing protocols for 3-D printed microfluidics using SLA-printing and resin. In addition, the authors demonstrate functional material characteristics and application for static PCR amplification. Overall, the paper is well written with good background detail and relevant literature review.

Minor Revisions Suggested:

1. Defining the post-processing steps used in a 3-D printed device is useful but building standards/controls for those processing methods is really what is needed to extend this work beyond the custom chip and application described. Maybe add a comment or two about how this could be or was addressed so that it advances 3-D DLP-SLA printing for the field.

2. Consider adding a scale bar on Figure 4.

3. The three-chamber chip design reduced bubbles (downfall of any microfluidic device) but it was unclear if all three chambers hold 15 uL and are used in the PCR reaction or only the central chamber. Please clarify.

4. Line 284, I believe Figure 5, (d, f) is mislabeled and should be (a, d).

5. Line 287, Figure 5 legend I believe the panels are mislabeled here as well. Shouldn’t it be (a,d) (b,e) and (c,f) with no i panel.

6. Line 311 should reference Figure 7, currently there is an error message.

7. In Figure 7, the gel band alignments and the measurement of the 75 bp amplicon don’t always seem to line up well. I would suggest removing the mfg standard ladder image and label the bp bands on your agarose gel within the figure. Furthermore in panel c, it appears that the gel conditions changed (% agarose?, if so please also indicate in methods or legend) and the largest marker bands are no longer present within the gel but it actually the only one that does align with the marker image at left side of the panel.

6. PLOS authors have the option to publish the peer review history of their article (what does this mean?). If published, this will include your full peer review and any attached files.

Reviewer #1: No

Reviewer #2: No

---

## [Author Response · Author response to Decision Letter 0]

27 May 2020

Part of the present text is mentioned in the uploaded document: Response to Reviewers

Dear Reviewers/Editors,

we would like to thank you for your time to review the manuscript and the positive evaluation of our work. In general, we consider that the points made by the reviewers are correct and the revision work made to address them improved the manuscript. In the following paragraphs we quote each comment and explain how the revised version of the manuscript is modified around it. We further highlight the updates made in regards with data availability and according to PLOS requirements.

Reviewer #1

1. “The miniaturized PCR device was claimed to be a microfluidic device. However, there was very little fluidic control demonstrated. The device looks more like a microwell to the reviewer. The authors shall specify more on the fluidic control, e.g. how was the PCR reagents delivered? Ideally, a better device design with fluidic control shall be demonstrated.”

The reviewer makes a good point. We have changed the terminology to refer to a microfluidic chamber, and we discuss the potential for more complex microfluidic devices. In lines 146-149, it was further explained that the microchamber architecture was developed to implement stationary PCR thermocycling which is more reliable compared to the error prone flow-through method when employing resin 3D printed chips as shown in the work of Park and Park (J. Park and H. Park, “Thermal cycling characteristics of a 3D-printed serpentine microchannel for DNA amplification by polymerase chain reaction,” Sensors and Actuators A: Physical, vol. 268, pp. 183-187, 2017). In this way it was possible to characterise PCR inhibition, exclusively owed to material interactions rather than poor temperature control. In the case of flow-through PCR, poor fluidic control emerging from variable deviation levels of printed channel dimensions can result in poor temperature control that would lead to false interpretations of material PCR compatibility.

2. "A related question is the level of complexity that DLP SLA printing would allow. The two-stage dip coating of the hydrophobic layer on chip may increase the production period and difficulty."

Complexity of channels achievable with the proposed methods is based on the melt viscosity of the wax, which determines the backpressure required for wax removal and feasibility depends on the bonding strength between the chip and the cap. In lines 354-356 it is mentioned that, the water-like melt viscosity of the low melting point paraffin wax, in combination with the unibody microchannel formed can deliver more complex 2.5D channels, such as heat exchanging architectures found in flow-through thermocycling setups. Although, such a geometry was not required in the present study, as it focuses specifically on the optimization of material properties to enable higher throughput platforms in the future. In regards with the second aspect of this comment, the coating processes for fluorination and glass coating lasted 2 hours and 1 hours respectively, as described in section 3.3 (line 368). These processes could be used for batches of chips, which was mentioned in the manuscript (line 375-376).

3. "The authors claimed that “the system is qPCR ready by integration with a fluorescent microscope and qPCR DNA stains. However, this was not demonstrated experimentally in the manuscript."

The text has been amended. To highlight the potential of the proposed method of thermocycling for 3D printed chips, the following statement was added in the conclusions: “From a system point-of-view, the proposed method of thermocycling shows promise for portability and modular integration of functionalities for diagnostic or research applications that utilize nucleic acid amplification technology.”

Reviewer #2: 

1. "Defining the post-processing steps used in a 3-D printed device is useful but building standards/controls for those processing methods is really what is needed to extend this work beyond the custom chip and application described. Maybe add a comment or two about how this could be or was addressed so that it advances 3-D DLP-SLA printing for the field."

Within this paper we present well defined protocols for the processing methods, which can be adopted by researchers more broadly as the first stage towards standardised methods. Extending the work beyond prototyping volumes is largely a commercial decision, based on the price per chip. We have added to the discussion (lines 454-458-) and conclusions (lines 470-472) to consider this point. Further to this comment, and in regards with the creation of standards/controls for the post-processing methods presented, a new section has been added (2.2. Characterisation of printed resin interference with PCR). This section describes a preliminary experiment, that indirectly demonstrates that prolonged UV curing and hot water baths removes PCR inhibiting substances on the prints, facilitating the reaction in low SVR architectures (fluidics and milli fluidics). This method of post-processing was then adopted for the high SVR printed chip geometry, showing that although PCR inhibiting residuals are removed, inhibition due to adsorption becomes dominant in high SVR non-treated prints. This conclusion is drawn, since only hydrophobically modified chips produced amplicons consistent with the controls.

2. "Consider adding a scale bar on Figure 4."

A scale bar was added, derived from the 3D CAD model assembly of the PCR thermocycler. Figure 4 was changed to Figure 5.

3. "The three-chamber chip design reduced bubbles (downfall of any microfluidic device) but it was unclear if all three chambers hold 15 uL and are used in the PCR reaction or only the central chamber. Please clarify."

It was clarified that all three chambers hold 15 μL of PCR mixture, with each chamber accommodating approximately 5 μL (lines 163-165). Temperature sensing was performed only on the central chamber with the sensing element in contact with the fluid.

4. "Line 284, I believe Figure 5, (d, f) is mislabeled and should be (a, d)."

Figure 5 was changed to Figure 6. The figure was relabelled and the referencing in the text was updated.

5. "Line 287, Figure 5 legend I believe the panels are mislabeled here as well. Shouldn’t it be (a,d) (b,e) and (c,f) with no i panel."

Figure 5 was changed to Figure 6. The figure was relabelled and the referencing in the text was updated.

6. "Line 311 should reference Figure 7, currently there is an error message."

Cross-references were updated.

7. "In Figure 7, the gel band alignments and the measurement of the 75 bp amplicon don’t always seem to line up well. I would suggest removing the mfg standard ladder image and label the bp bands on your agarose gel within the figure. Furthermore in panel c, it appears that the gel conditions changed (% agarose?, if so please also indicate in methods or legend) and the largest marker bands are no longer present within the gel but it actually the only one that does align with the marker image at left side of the panel."

Figure 7 was changed to Figure 8. The DNA ladder bands on the gel images were tagged individually. The text and the caption of the figure were further updated by clarifying that the gel representing the reactions performed on a non-treated chip (c panel) was 2% similar to a and b panels, but run for 20 minutes, instead of 45 for the modified printed chips that allowed for full ladder separation. This was for visualizing the faint amplicon generated on the non-treated chip before it was fully absorbed in the agarose gel at the end of the 45-minute run. Due to the early interruption of the agarose gel electrophoresis, the DNA ladder bands in Figure 8 (c), c are not fully separated but appear in groups for 766-100 and 75-25 base pair. This was confirmed by a few-minute additional rerun, which was added as an inset in Figure 8, (c) and is provided in the supporting information (S1_raw_images). However, the image comprehensively conveys the message of intense PCR inhibition owed to the reactivity of the non-treated printed polymer that creates suboptimal conditions, thereby disrupting the outcome of the reaction, providing a false negative.

Further changes according to PLOS requirments are summarized below.

According to PLOS style requirements, we formatted the manuscript according to the online template.

According to PLOS requirements for data availability, we have uploaded the raw data of the paper, including the blot/gel image data according to PLOS requirements, in the data repository of Newcastle university. Separate captions have also been added in the supporting information files. The supplementary information description and individual DOIs are mentioned below, as well as within the manuscript, under each figure and in the related section.

The supporting information files added and their DOIs are summarized below: 

• S1A_raw_images: Raw agarose gel electrophoresis images for the mock PCR reactions presented in 2.2 and on-chip reactions presented in 3.5, Figure 5. (10.25405/data.ncl.12320501)

• S1B_Fig_8_Amplicon_band¬_intensity_plots: Amplicon band intensity profiles of the on-chip reactions presented in 3.5, Figure 8. (10.25405/data.ncl.12320396)

• S2A_Fig_6_SEM_contact_angle: Raw SEM images and water contact angle visualizations of printed and functionalized disk specimens presented in 3.3, Figure 6. (10.25405/data.ncl.12320837)

• S2B_contact_angle: Contact angle values for all functionalized and non-functionalized printed disk specimens presented in 3.3, Figure 6. (10.25405/data.ncl.12320849)

• S3_Fig_6_SEM_EDS: Elemental analysis of the surface of printed and functionalized disk specimens presented in 3.3, Figure 6. (10.25405/data.ncl.12320861)

• S4_ Fig_7_2_step_PCR-temperature_profile: Temperature profile of the IR PCR system presented in 3.4, Figure 7 for a glass-coated 3D-printed chip ran a two-step PCR thermocycling protocol (10.25405/data.ncl.12320864).

An ORDID ID was created for the corresponding author and each author was linked to an institution within the manuscript. 

All figures are now referenced within the text and are presented just after the text that cites them. Figure captions within the manuscript were also formatted according to guidelines. 

We hope that we successfully addressed the reviewers’ comments and clarified the presented data, giving a good insight into the versatile capabilities of high-resolution 3D-printing, specifically for microfluidics. We hope that the manuscript now meets the requirements for publication in PLOS one.

Yours sincerely,

Charalampos Tzivelekis

---

## [Decision Letter · Decision Letter 1]

3 Sep 2020

PONE-D-20-09489R1

Fabrication routes via projection stereolithography for 3D-printing of microfluidics for nucleic acid amplification

PLOS ONE

Dear Dr. Tzivelekis,

Thank you for submitting your manuscript to PLOS ONE. After careful consideration, we feel that it has merit but does not fully meet PLOS ONE’s publication criteria as it currently stands. Therefore, we invite you to submit a revised version of the manuscript that addresses the points raised during the review process.

Please address the points raised by reviewer 3.

We look forward to receiving your revised manuscript.

Kind regards,

Andreas Offenhausser

Academic Editor

PLOS ONE

Journal Requirements:

Additional Editor Comments (if provided):

Reviewers' comments:

Reviewer's Responses to Questions

**Comments to the Author**

1. If the authors have adequately addressed your comments raised in a previous round of review and you feel that this manuscript is now acceptable for publication, you may indicate that here to bypass the “Comments to the Author” section, enter your conflict of interest statement in the “Confidential to Editor” section, and submit your "Accept" recommendation.

Reviewer #1: All comments have been addressed

Reviewer #3: (No Response)

2. Is the manuscript technically sound, and do the data support the conclusions?

Reviewer #1: Yes

Reviewer #3: Yes

3. Has the statistical analysis been performed appropriately and rigorously? 

Reviewer #1: Yes

Reviewer #3: Yes

4. Have the authors made all data underlying the findings in their manuscript fully available?

Reviewer #1: Yes

Reviewer #3: Yes

5. Is the manuscript presented in an intelligible fashion and written in standard English?

Reviewer #1: Yes

Reviewer #3: Yes

6. Review Comments to the Author

Reviewer #1: (No Response)

Reviewer #3: The original manuscript convincingly demonstrated the need for silanization of 3D printed PCR wells to obtain useful yields of PCR amplicons. The revised manuscript appropriately answered the reviewers’ questions.

A few other minor questions could be answered by the authors in order to strengthen the revised manuscript further:

1. One of the problems with SLA 3D printing is the volume shrinkage of the material used upon curing, which could induce high internal stress causing the deformation of material, and may eventually break the material. Did the authors observe or envision any such issues with the material they used? Did curing deformation affect the precision of the printed model?

2. Many resin materials with low viscosity have problems with biocompatibility, perhaps due to the residual photoinitiator or the degree of crosslinking of the material due to low viscosity. Can the authors comment on how the post-processing treatments and silane coupling agents affected biocompatibility?

3. Line 232, Figure 1 legend, change “positioning it” to “positioned”.

4. Line 315, Figure legend 4, add “for” to read it as “basis for further exploration”.

5. Line 423, Figure legend 7, change “denature” to “denaturation”.

7. PLOS authors have the option to publish the peer review history of their article (what does this mean?). If published, this will include your full peer review and any attached files.

Reviewer #1: No

Reviewer #3: No

---

## [Author Response · Author response to Decision Letter 1]

14 Sep 2020

In the following paragraphs we quote the comments made by reviewer #3 and provide a point-by-point response, where we discuss the points and explain how these are addressed in the manuscript.

Reviewer #3: 

"The original manuscript convincingly demonstrated the need for silanization of 3D printed PCR wells to obtain useful yields of PCR amplicons. The revised manuscript appropriately answered the reviewers’ questions. A few other minor questions could be answered by the authors in order to strengthen the revised manuscript further:"

"1. One of the problems with SLA 3D printing is the volume shrinkage of the material used upon curing, which could induce high internal stress causing the deformation of material, and may eventually break the material. Did the authors observe or envision any such issues with the material they used? Did curing deformation affect the precision of the printed model?"

We have previously experienced deformations in microfluidic prints induced by over-curing with various resins. These usually result in flatness issues, especially for large print geometries. In this project, the utilized resin is commercially developed for high-definition prints of high thermal stability (e.g. moulds for injection moulding) and displays low deformation. By following the manufacturer’s instruction (Formlabs, 2016) for post-UV curing at approximately 80 mW/cm2*, we observed no distortion, at a level that would impair the device performance. The chips were designed to accommodate 15 μL fluid volumes, which were precisely loaded using a mechanical pipette. The reaction chamber withstood the pressure built during PCR thermocycling, verifying the good bonding quality, while the material remained rigid with no signs of deterioration. Flatness issues were not observed, potentially due to the small chip area and thickness which allowed a level of flexibility.

* chips were cured in a custom-built UV oven (355 nm, 0.003- 0.3 W/cm2, resolution 0.003 W/cm2)

This point was addressed in the manuscript:

In 2.2, 

line 186-189: “The utilized resin is commercially developed for high-definition prints of high thermal stability and minimum deformation and its main application is 3D printing of thermoforming tools. Layer exposure time was selected at 2 ± 0.1 seconds, so that optimal feature definition is achieved, as it was visually observed (Fig 1, 6)”, 

line 191-192: “…according to the manufacturer instructions to achieve heat deflection temperature higher than 200°C”, 

in 3.2, 

line 382-386: “By following the manufacturer instruction for post-UV curing with an 80 mW/cm2 light source, no distortion was observed at a level that would impair the performance of the printed device. The chips were designed to accommodate 15 μL fluid volumes, which were precisely loaded using a mechanical pipette. No flatness issues were observed, while the small chip area and thickness allowed for a level of flexibility.”, 

in 3.4, 

line 423-425: “In terms of the material performance in thermocycling, the reaction chamber withstood the pressure built during PCR and the material remained rigid with no visual signs of deterioration.”

"2. Many resin materials with low viscosity have problems with biocompatibility, perhaps due to the residual photoinitiator or the degree of crosslinking of the material due to low viscosity. Can the authors comment on how the post-processing treatments and silane coupling agents affected biocompatibility?"

Biocompatibility in the wider sense was not part of this study. The focus was on developing a material suitable for nucleic acid amplification, in a sense that it does not interfere with the amplification process. In this context, the post processing treatments with ultraviolet (UV) curing and washing aim to reduce the amount of residuals (e.g. monomers, oligomers, photoinitator) which can be strong PCR inhibitors (Inhibitory effect of common microfluidic materials on PCR outcome, 2012; Compatibility analysis of 3D printer resin for biological applications, 2016; Inhibition and Facilitation of Nucleic Acid Amplification, 1997). UV curing achieves reduction of residuals through post-polymerization (i.e. consumption of the residuals in the polymer chain), while washing achieves reduction through removal with solvents of high monomer solubility (including water). Silane coupling agents created a hydrophobic passivated surface, reducing the potential for adsorption of components vital to the reaction, such as Taq polymerase (Biochemical analysis and optimization of inhibition and adsorption phenomena in glass-silicon PCR-chips, 2003; Relief of Amplification Inhibition in PCR with Bovine Serum Albumin or T4 Gene 32 Protein, 1996). Considering similar literature and our obtained results, the enzyme is the main adsorbed PCR component. The proposed hydrophobic treatments are appropriate for reducing biological adsorption in 3D printed microfluidics.

This comment was addressed in the manuscript by adding the following statements: 

In 3.1, 

line 358-361: “UV curing reduced PCR inhibiting residuals by inducing their consumption into the polymerized material and increasing the degree of cross-linking. Water and isopropanol further dissolve print residuals to remove them from the printed surface.”,

in 3.5, 

line 476-480: “Silane treatments exploit the surface reactivity to create a hydrophobic passivated surface, which significantly limits the adsorption interface and level of reactivity. PCR components could not be adsorbed on the hydrophobic surface and the reaction was facilitated. Considering similar literature results, enzyme is the main adsorbed PCR component (Biochemical analysis and optimization of inhibition and adsorption phenomena in glass-silicon PCR-chips, 2003; Relief of Amplification Inhibition in PCR with Bovine Serum Albumin or T4 Gene 32 Protein, 1996). The proposed hydrophobic treatments are appropriate for reducing biological adsorption in 3D printed microfluidics.”

"3. Line 232, Figure 1 legend, change “positioning it” to “positioned”."

"4. Line 315, Figure legend 4, add “for” to read it as “basis for further exploration”."

"5. Line 423, Figure legend 7, change “denature” to “denaturation”."

The figure legends were updated.

We think that the changes further clarify the presented data and highlight the capability of stereolithorgaphy 3D printing to improve the efficiency of microfluidics rapid prototyping even for challenging applications, under appropriate processing protocols and stategies.

---

## [Editor Report · Decision Letter 2]

23 Sep 2020

Fabrication routes via projection stereolithography for 3D-printing of microfluidic geometries for nucleic acid amplification

PONE-D-20-09489R2

Dear Dr. Tzivelekis,

We’re pleased to inform you that your manuscript has been judged scientifically suitable for publication and will be formally accepted for publication once it meets all outstanding technical requirements.

Kind regards,

Andreas Offenhausser

Academic Editor

PLOS ONE
---

## [Editor Report · Acceptance letter]

9 Oct 2020

PONE-D-20-09489R2 

Fabrication routes *via* projection stereolithography for 3D-printing of microfluidic geometries for nucleic acid amplification 

Dear Dr. Tzivelekis:

I'm pleased to inform you that your manuscript has been deemed suitable for publication in PLOS ONE. Congratulations! Your manuscript is now with our production department. 

Kind regards, 

on behalf of

Dr. Andreas Offenhausser 

Academic Editor

PLOS ONE